# A SOLUTION TO CHINA COMPETITIVE POKER USING DEEP LEARNING

## ABSTRACT

Recently, deep neural networks have achieved superhuman performance in various games such as Go, chess and Shogi. Compared to Go, China Competitive Poker, also known as Dou dizhu, is a type of imperfect information game, including hidden information, randomness, multi-agent cooperation and competition. It has become widespread and is now a national game in China. We introduce an approach to play China Competitive Poker using Convolutional Neural Network (CNN) to predict actions. This network is trained by supervised learning from human game records. Without any search, the network already beats the best AI program by a large margin, and also beats the best human amateur players in duplicate mode.

## 1 INTRODUCTION

China Competitive Poker (CCP) is a poker game known as easy to learning but hard to master, requiring the mathematical and strategic thinking as well as carefully planned execution. CCP is played among three people with one pack of cards, including two jokers. After bid, one player would be the "Landlord (or Declarer)", the other players are "Peasant (or Defenders)", they are a team competing against the "Landlord". Starting from the Landlord, players take turns to play sets/groups of cards. The objective of the game is to be the first player to play out all the cards. More details about CCP are described on Wikipedia (2018b).

CCP is certified competitive sport by General Administrator of Sport of China. Dou dizhu has 15.4 million DAU (Daily Active User) in Tencent game platform, which is the largest online platform in China(Baijiahao, 2018). Compared to Go, CCP is a kind of imperfect information game, starts from a random state. A player gets different cards each game, does not know the cards distribution of the other two players. Like bridge, Peasants must coordinate with each other, otherwise they could hardly beat the Landlord. CCP is a good problem for game AI. We choose CNN to solve the problem in CCP due to the following reasons: First, CNN has achieved superhuman performance in perfect information games. Second, there is semi-translational invariance in CCP, e.g. there are two sets of cards in the same category but with different ranks (like "34567" and "45678", see more information in section 3), if we add each card's rank, "34567" become "45678", this is translational invariance. The player can play out "45678" after another one played out "34567", but it is illegal if we swap the order, this is the reason for "semi".

There is no AI about CCP using deep neural network so far. It is still to be proved that the network can output a proper action with an imperfect information input in CCP. The same factor between CCP and Go is blurred when making a decision. It is very hard to evaluate the value of alternative actions in Go, also CCP. As there are teammates in CCP, one problem is to teach the network to cooperate, e.g. how to help teammate become the first one to play out all cards, while his cards is not good enough. Another problem we are interested in is inference, whether the network is able to analyze like human experts' logic. The DeepRocket, which is developed by us, gets the state of the art performance in CCP. We prove that the network can learn cooperate and inference in imperfect information game.

In section 2, we introduce related work about CCP. In section 3, introduce some notations in CCP, while is used through this paper. We explain each component in section 4, including details about the policy network and the kicker network. We show how to prepare the experiment, the results

of competing with the best previous AI, and top human amateur players in section 5. Finally, the summary and the problems to be improved is shown in section 6.

## 2 RELATED WORK

As so far, Libratus and DeepStack are the best AI programs in Texas Hold'em, which is also imperfect information game, developed by Carnegie Mellon University and University of Alberta respectively(Brown & Sandholm, 2017; Moravčík et al., 2017). Only DeepStack use the deep neural network, and the problem is both of them can only be used in the one-to-one mode.

University Computer Games Championship & National Computer Games Tournament is the largest AI competition in China. CCP is opened to not only universities but also the public. More than ten teams participate the tournament each year, all are based on logic code, rather than deep neural network, the champion cannot beat the human amateur players until now.

## 3 NOTATIONS

This section introduces the notations in CCP, as well as the notations we use throughout this paper.

The Specified Player: is one player we stand in his perspective.

Down Player: is the player who does action after the Specified Player.

Up Player: is the player who does action before the Specified Player.

Landlord: is the player who bid the highest score.

Down Peasant: is the player who does action after Landlord.

Up Peasant: is the player who does action before Landlord.

Public Cards: Each player get 17 cards. Landlord can get the last 3 cards, but have to show them to all players. These 3 cards are Public Cards.

Round: All players have done actions once sequentially, started from the specified player. "33;99;PASS" is a round information, it represents "33" is played out by the Specified Player, next "99" is played out by Down Player, finally, Up Player choose "PASS".

Category: All legal types of cards in CCP, like "33344" is legal while "33345" is not.

Main Group and Kicker Card: a solo or a pair is allowed to be played out with Trio. Trio is named as "Main Group", the solo or the pair is named as "Kicker Card". The rule is also used in Trio with Chain. For example, when "3334" or "33355" is played out, "333" is Main Group, "4" or "55" are Kicker Cards. Two solos or two pairs are played out with Four (A kind of category in CCP).

Set or Group: is one kind of categories, e.g. "334567" is divided into two sets, "3" and "34567".

Active Mode: A player can play any set available, if the following occurs:

- Landlord first play in the game;
- Both the last two actions are "PASS".

Passive Mode: Except Active Mode, subsequent player must play a set of the same category but higher rank (bomb and rocket are not included).

Duplicate Mode: Analogous to Duplicate Bridge(Wikipedia, 2018a) (also known as Tournament Bridge), it provides a more robust evaluation for players' performance. More details is provided in Appendix B.

## 4 DEEPROCKET FRAMEWORK

The system named DeepRocket consists of three parts: the Bid Module, the Policy Network and the Kicker Network. When game starts, the Bid Module is called in order to make DeepRocket output

score, if necessary. The Policy Network is called before DeepRocket will do action, the output may contain Kicker Card type (solo or pair) in some labels. If there is a Kicker card need to be predicted, then the Kicker Network is called, as shown in Figure 1 and Figure 2.

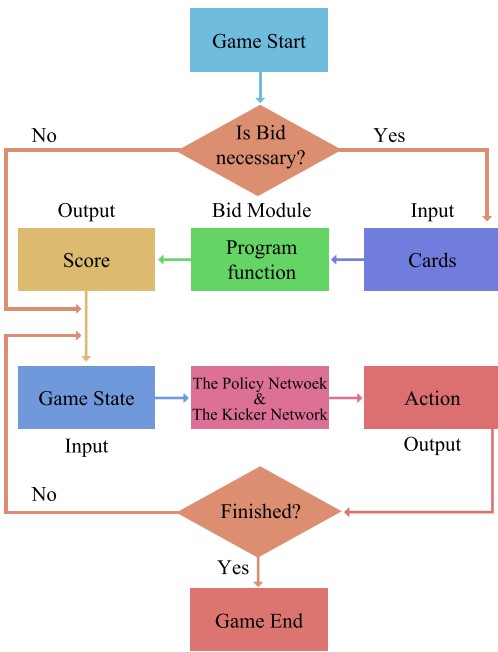

Figure 1: DeepRocket game flow.

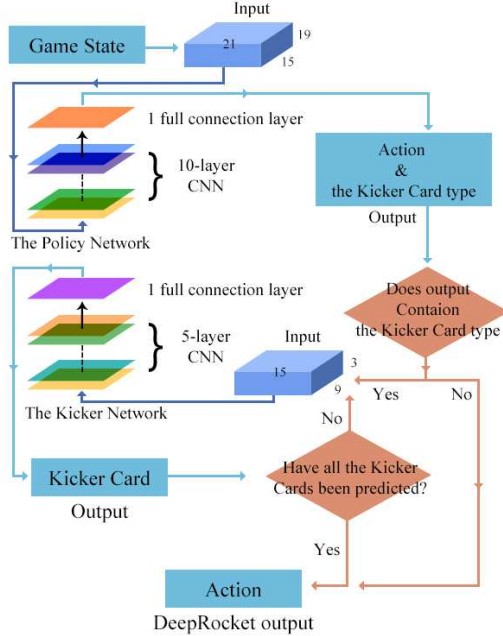

Figure 2: The Policy Network and The Kicker Network work flow.

Table 1: State-action pairs divided from the game record

| SAMPLE INDEX | GAME STATE | ACTION (LABEL) |
|---|---|---|
| 1 | 0,33; | 1,55 |
| 2 | 0,33;1,55;2,66;0,77; | 1,AA |
| 3 | 0,33;1,55;2,66;0,77;1,AA | 1,6 |
| 4 | 0,33;1,55;2,66;0,77;1,AA;1,6;2,T;0,J; | 1,K |
| ... | ... | ... |

## 4.1 THE BID MODULE

First of all, the game need to make sure who is Landlord by bid after players get cards, so we design the module for bid. The key factors of bid is cards. The Bid module is based on logic code. Like human experts, the output of module depends on control cards (like "A", "2", and "jokers") and tidiness (that means less sets combined by cards). The module have been used on many online platform in business in China, and it is proved that works well. Bid is not used in Duplicate Mode, also in test, we do not introduce more details and focus on other components of DeepRocket.

## 4.2 THE POLICY NETWORK

Like AlphaGo(Maddison et al., 2015; Silver et al., 2016), we trained the Policy Network to predict human expert actions using supervised learning. The Policy Network consists of 10-layer CNN and 1 full connection layer, use Relu as activation function. A final softmax layer outputs probability distribution over all legal action $a$. The input represents current game state. The Policy Network is trained on randomly sampled state-action pairs $(s, a)$, using stochastic gradient ascent to maximize the likelihood of the human expert action. We use 8 million game records for training the Policy Network. One record represents a complete game, and can be divided into many state-action pairs, from several rounds to more than twenty rounds, depending on length of the records. Here is a example, see Appendix A for more details about the game record.

Cards:4456777889JKKAA2B;335567899TTJJKAA2;4456689TTJQQQK22S;33Q

Game process:0,33; 1,55; 2,66; 0,77; 1,AA; 1,6; 2,T; 0,J; 1,K; 0,2; 2,S; 2,44; 0,KK; 2,22; 2,89TJQK; 2,QQ; 0,AA; 0,56789; 1,789TJ; 1,3; 2,5

If we want to learn Down Peasant, the record should be divided into the state-action pairs shown in Table 1 (the order is shuffled).

The input to the Policy Network is a $15 \times 19 \times 21$ 3-dimensional binary tensor (we name the corresponding dimension X, Y, and Z respectively). Each dimension along X-axis represents the ranks of cards, from 3 to big joker. Y-axis represents the number of each rank (from 1 to 4), and features in CCP, such as solo, pair, trio, etc. Z-axis represents the sequential information of each rounds, this design learned from AlphaGo, make the variable length to fixed length in game. Finally, the most recent 6 rounds are chosen. $S_{i,j,k}$ standards the binary feature for current state of the game. Details are shown in table 2.

512 filters are the most suitable after models are tested. 10-layers CNN achieved the best performance after repeated testing, different strides are used in each layer.

The Policy Network outputs 309 action probabilities after we add the Kicker Network to Deep-Rocket, See table 3. It is noteworthy that the maximum number of cards is 20, it restricts the longest length of category.

## 4.3 THE KICKER NETWORK

A combination of the Bid Module and the Policy Network suffices to play a game in principle. However, there is a problem dealing with Kicker Cards. If there are $n$ choice for the Main Group and $m$ for the Kicker Cards, then there are $n \times m$ available actions in total. In some other case, if the player has trio with chain (e.g. "333444" is a trio with chain), the length is $i$ (the length

Table 2: The meaning of Z-axis

| PLANES | MEANING | |
|---|---|---|
| 1 | All played out cards before last 6 rounds. | |
| 2-4 | The played out cards in the sixth last round. | Three planes represent the cards of three players in this round respectively, e.g. No 17 plane represents the cards of the Specified Player, No 18 plane represents the cards of Down Player, and No 19 plane represents the card of Up Player in the most recent round. |
| 5-7 | The played out cards in the fifth last round. | |
| 8-10 | The played out cards in the fourth last round. | |
| 11-13 | The played out cards in the third last round. | |
| 14-16 | The played out cards in the penultimate round. | |
| 17-19 | The played out cards in the most recent round. | |
| 20 | All cards that have not been seen yet. | |
| 21 | All cards in hand. | |

Table 3: The output's composition

| PRIMAL | WITH(+) KICKER | CHAIN | QUANTITY | DESCRIPTION |
|---|---|---|---|---|
| Solo | No | No | 15 | There are two types for red and black jokers. |
| Solo | No | Yes | 36 | It has different length, from 5 (the shortest) to 12 (the longest). |
| Pair | No | No | 13 | Both jokers are rocket, not a pair. |
| Pair | No | Yes | 52 | It has different length, from 3 (the shortest) to 10 (the longest). |
| Trio | No | No | 13 | As much as the same with Pair without chain. |
| Trio | No | Yes | 45 | It has different length, from 2 (the shortest) to 6 (the longest). |
| Trio | Solo | No | 13 | E.g. "3334" and "3335" are the same type for the Policy Network. |
| Trio | Solo | Yes | 38 | It has different length, from 2 (the shortest) to 5 (the longest). |
| Trio | Pair | No | 13 | E.g. "33388" and "333KK" are the same type for the Policy Network. |
| Trio | Pair | Yes | 30 | It has different length, from 2 (the shortest) to 4 (the longest). |
| Four | Dual solo | No | 13 | Four cards of the same rank with two distinct individual cards as the Kicker. |
| Four | Dual pair | No | 13 | Four cards of the same rank with two sets of pair as the kicker. |
| Bomb | No | No | 13 | Four cards of the same rank without the Kicker. |
| Rocket | No | No | 1 | Only 1 type, both jokers. |
| Pass | No | No | 1 | - |

of "333444" is 2), and $j$ for available Kicker Cards, then there are $C_j^i$ choice in total. For example, "333444555666789J" is a legal action, but the sample is hardly found in even 8 million game

Table 4: The kicker network's input channel

| CHANNEL/PLANE | IMPLICATION |
| --- | --- |
| 1 | The information about kicker type, role, remaining cards, legal solo and pair. |
| 2 | Record information about the game, Main Group and Kicker length. |
| 3 | Remaining cards, legal cards and other features. |

records. Furthermore, Main Group with different Kicker Cards is labeled as different actions, and thereby a combinatorial number of different actions to be predicted by the Policy Network.

In order to address this problem, we add an extra network for predicting the Kicker Card. The Policy Network predict the Main Group and Kicker Cards type, like solo or pair, then the Kicker Network predict the kicker cards. This make the Policy Network only output 309 probabilities rather than thousands. For example, "333444555666789J" and "333444555666789Q" are the same output in the Policy Network (outputs "333444555666" and solo kicker cards) while "789J" or "789Q" is predicted by the Kicker Network.

The input to the Kicker Network, which include the remaining cards and the output of the Policy Network is a 15×9×3 3-dimensional binary tensor. Each dimension along X-axis represents the cards, is the same as the Policy Network. Table 4 shows the details about Y-axis and Z-axis. The Kicker Network has 28 outputs, include 15 kinds of solo and 13 kinds of pair.

The Kicker Network consists of 5-layer CNN and 1 fully connection layer and outputs the probabilities of Kicker Cards. The Kicker Network outputs one type kicker each time. If the Policy Network predict "333444" and 2 solo Kicker Cards, the Kicker Network should be called twice.

# 5 EXPERIMENTS

## 5.1 EXPERIMENT SETUP

We got 8 million game records, and divided them into about 80 million state-action pairs. 90% of them were train set, while 10% were test set, then made them to be the networks' inputs. Finally, the inputs were stored to hard disk using TFRecords. It is not only convenient to change the parameters of the networks, but also make the train faster. It spend about 20 hours for training the Policy Network each time. The batch size of the Policy Network is 256. The Policy Network predict human expert actions with an accuracy of 86-88% on the test set. It takes 0.01-0.02s to compute output of the Policy Network, using the server which use i7-7900X CPU, NVIDIA 1080Ti GPU and Ubuntu 16.04 server system.

The Kicker Network is also trained by supervised learning, all samples come from the 8 million game records. The Kicker Network reach 90% accuracy in test set after training, even better than the Policy Network.

We changed the layers, filters and other parameters and got many models. We also used the Duplicate Mode to test models, compared Landlords' cumulative scores in 10000 games, and chose the best one.

## 5.2 COMPARISON WITH THE CURRENT BEST AI

MicroWe won the championship in University Computer Games Championship & National Computer Games Tournament 2017. MicroWe was the best CCP artificial intelligence before the appearance of DeepRocket.

Like Figure 3 and Figure 4 shows, we totally tested 50000 games, each iteration represents 5000 games. What need special explanation is we sent 20 cards into a position directly, and assign it to be Landlord. Therefore Landlord's win rate are lower than usual. "DR VS MW" means that DeepRocket is Landlord while MicroWe is Peasants.

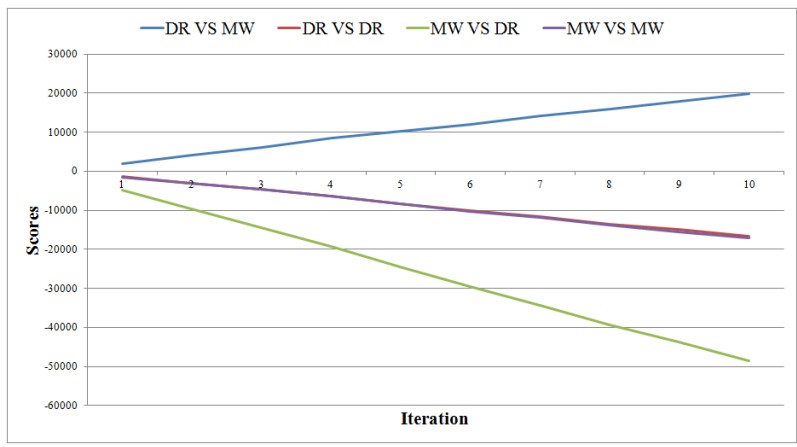

Figure 3: Match result of DeepRocket (DR) and MicroWe (MW)

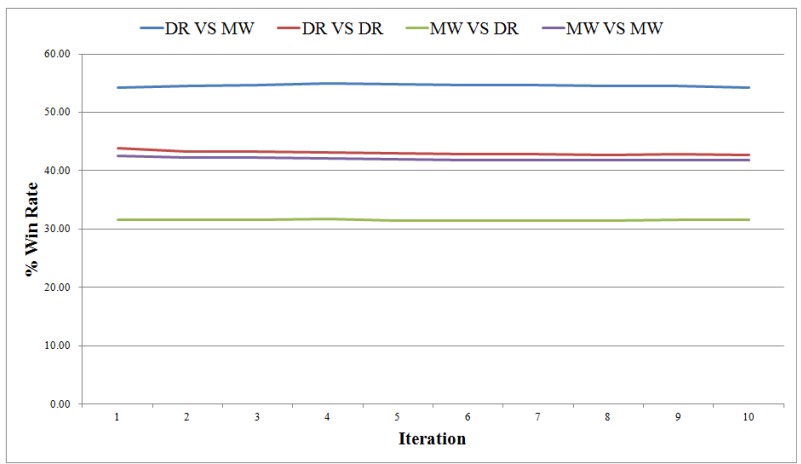

Figure 4: Win rate between different AI

We get conclusion that DeepRocket is obviously better than MicroWe, which is the best AI in CCP before DeepRocket.

## 5.3 COMPARED WITH HUMAN EXPERTS

A test match was hold and four top amateur players were invited. Duplicate Mode was used in the match, 10 games in total. Finally, DeepRocket beat the human team with a score of 30 to 24, it means DeepRocket got 30 scores as Landlord, while human players got 24 scores as Landlord in 10 games. More details see Appendix C.

Participant introduction:

Huaou Xu: famous player in Texas Hold'em and CCP, participated as Landlord in 5 of 10 games.

Zongchao Cheng: famous player in CCP, participated as Landlord in 5 of 10 games.

Lanzhou Zheng: famous player in Texas Hold'em and CCP, participated as Up Peasant in games.

Guofeng Xie: won the CCP champion in FGG games, participated as Down Peasant in games.

## 5.4 Cooperation and inference

Another problem we'd like to research is cooperation between agents. In a DeepRocket's game record, we found a typical example (see Appendix A for more details about game record):

Cards: 4456777889JKKAA2B;335567899TTJJKAA2;4456689TTJQQQK22S;33Q;

Game process: 0,33;1,55;2,66;0,77;1,AA;1,6;2,T;0,J;1,K;0,2;2,S;2,44;0,KK;2,22;2,89TJQK;2,QQ; 0,AA;0,56789;1,789TJ;**1,3**;2,5;

The snapshot we analysis is after Down Peasant played out "789TJ", which is highlight in bold in game process. The remaining cards are:

Landlord: "448QB";

Down Peasant: "339TJ2";

Up Peasant: "5".

Down Peasant was in Active Mode, that means any available set or group can be played out. The DeepRocket played out "3" to help the partner won the game, also himself.

The last issue we are concerned about is inference. We found a typical example also:

Cards: 33345578TTJKKA22S;34566789TQQQKKAA2;4456678999TJJQA2B;78J;

Game process: 0,345678;1,56789T;0,6789TJ;0,QQQKK;0,AA;2,22;2,55;2,3334;2,TT;**2,A**;0,2;

The key action is highlighted in bold also. The remaining cards are:

Landlord: "2";

Down Peasant: "44699JJQA2B";

Up Peasant: "78JKKAS".

Despite Peasants lost at last, it is a good choice that the Up Peasant played out "A". Landlord had only 1 card left, Up Peasant had 5 solo cards ("7, 8, J, A, S"). "A" is a human expert logic action, and DeepRocket learned from human.

## 6 Discussion

We have shown that CNN is able to predict actions in CCP, cooperate with teammate, achieve the same level as top human amateur players, even higher, without any MCTS.

Apart from this, there is still lots of work to be done. The first issue is reinforcement. It does not work well when we directly transplant the method of AlphaGo into CCP, and finally, we found a effective method to make the network better. The second issue is Monte Carlo search or MCTS. Human expert deduce the accurate cards distribution of the other players. In such game, human expert depends on inference. E.g. Landlord played out "3" at first, and has one "K" in hand. Down Peasant chose "PASS", and Up Peasant played out "K". Human expert deduces the maximum probability of "K" distribution is Down Peasant has two "K". Whitehouse et al. (2011) had introduced the method about MCTS in Mini Dou dizhu (also can be named Mini CCP), which only has 4 ranks and jokers. We have been working on effective methods for searching in CCP, and will publish them in future papers. Combination of the network and the method beat the only network easily.

DeepRocket can be improved in following aspects:

- Bid should be improved. Only 0, 1, 2, and 3 is legal in bid, 0 means the player does not want to be Landlord. Actually, 0 and 3 are most commonly used. We will try a deep neural network to train bid.

- We will try to train a model from random weight, like AlphaZero(Silver et al., 2017b;a).

- We will try to train a model with three outputs, which represent three roles respectively.

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

## A  DETAILS ABOUT GAME RECORD

Each number or letter represents a card's rank in game. Because the three of club and the three of diamond are the same in CCP, we do not distinguish Spade, Heart, Diamond, and Club in game record. Table 5 shows the meaning of letter in game record.

A game record is divided into two parts, as shown in Table 6: Cards and Game process.

Cards is composed of Landlord's cards, Down Peasant's cards, Up Peasant's cards, and the last 3 cards, sequentially. E.g. "335567899TTJJKAA2" are Down Peasant's cards. Game process is divided into many tuples by ";". Each tuple represents a action, the first number represents the role (0, 1, 2 represent Landlord, Down Peasant, Up Peasant respectively), the second part represents the played out cards. For instance, "0,33;" represents 0 (Landlord) played out "33", and then Down Peasant played out 55 ("1,55;"). Because CCP is played in order, "PASS" is ignored in game record.

Table 5: The meaning of letter in game record

| THE LETTER | THE MEANING | THE LETTER | THE MEANING |
|---|---|---|---|
| T | 10 | J | Jack |
| Q | Queen | K | King |
| A | Ace | S | Small Joker |
| B | Big Joker | - | - |

Table 6: Example for the game record

| Cards | 4456777889JKKAA2B;335567899TTJJKAA2;4456689TTJQQQK22S;33Q; |
|---|---|
| Game process | 0,33;1,55;2,66;0,77;1,AA;1,6;2,T;0,J;1,K;0,2;2,S;2,44;0,KK;2,22;2,89TJQK; 2,QQ;0,AA;0,56789;1,789TJ;1,3;2,5; |

## B  DUPLICATE MODE

The result of test are subject to fluctuations in CCP. The game of Go begin at the same state while CCP begin at random state, that means player get cards randomly. It need long time to distinguish who is the better player.

Duplicate mode can provide a more robust evaluation. First of all, duplicate mode need more than one table (usually two tables). Cards are dealt by referee, Landlord position is confirmed by referee also. Three players are a team, and use two tables: table 1 and table 2. One player is Landlord, Two AI are Peasants in table 1; AI is Landlord, two players are Peasants in table 2. The same role in two tables get the same cards dealt by referee. The team win if their Landlord's cumulative of scores is higher. Details are shown in Figure 5.

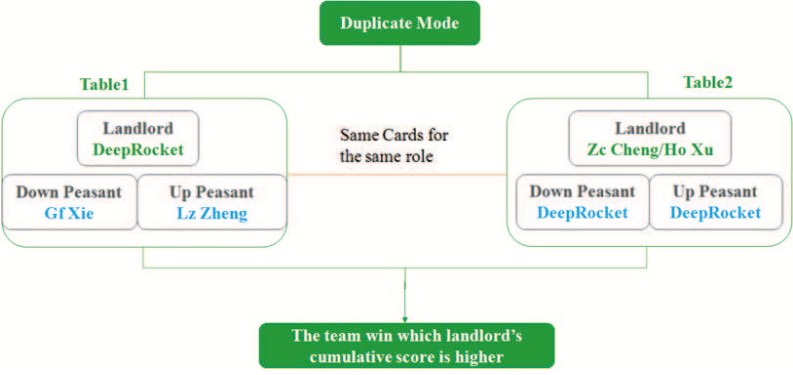

Figure 5: Duplicate mode

## C  TEST MATCH

We list the 10 results and games records for test match. The basic score is 3. It means Landlord get 6 scores, and Peasant get -3 scores if Landlord win without any bomb or rocket. "DP" represents the Down Peasant, "UP" represents the Up Peasant in table, "DR" represents the DeepRocket.

Table 7: The result of test match

| NUMBER | TABLE | LANDLORD | DP | UP | HUMAN | DR |
|---|---|---|---|---|---|---|
| 1 | 1 | DR | Gf Xie | Lz zheng | **6** | **12** |
| | 2 | Zc Cheng | DR | DR | | |
| 2 | 1 | DR | Gf Xie | Lz zheng | 12 | 12 |
| | 2 | Zc Cheng | DR | DR | | |
| 3 | 1 | DR | Gf Xie | Lz zheng | -12 | -12 |
| | 2 | Ho Xu | DR | DR | | |
| 4 | 1 | DR | Gf Xie | Lz zheng | 12 | 12 |
| | 2 | Ho Xu | DR | DR | | |
| 5 | 1 | DR | Gf Xie | Lz zheng | 6 | 6 |
| | 2 | Ho Xu | DR | DR | | |
| 6 | 1 | DR | Gf Xie | Lz zheng | -6 | -6 |
| | 2 | Ho Xu | DR | DR | | |
| 7 | 1 | DR | Gf Xie | Lz zheng | **12** | **-6** |
| | 2 | Ho Xu | DR | DR | | |
| 8 | 1 | DR | Gf Xie | Lz zheng | 12 | 12 |
| | 2 | Zc Cheng | DR | DR | | |
| 9 | 1 | DR | Gf Xie | Lz zheng | **-12** | **6** |
| | 2 | Zc Cheng | DR | DR | | |
| 10 | 1 | DR | Gf Xie | Lz zheng | -6 | -6 |
| | 2 | Zc Cheng | DR | DR | | |

Table 8: Initial cards of test match

| NUMBER | CARDS |
|---|---|
| 1 | 334566789JJQQKASB;34456789TTKAAA222;345577889TTJQQKK2;69J |
| 2 | 34445689TJQKKA22B;4555677889TJJQKA2;3336677899TTJQQK2;AAS |
| 3 | 356777899TQKAA222;3445557889TTJJKK2;334466689TJQQKASB;JQA |
| 4 | 346799TTTJJQKKK2B;366678899TJQQAA22;34445557788JQAA2S;35K |
| 5 | 3444789TTJQQKAA2B;3455667899JJJKA2S;355667788TQQKKA22;39T |
| 6 | 35577789JJJQKA22B;44566889TTTJQKK2S;3334567899TQQKAAA;462 |
| 7 | 678899TJQQKKAA22S;34455566779TTQA22;33345679TJJJQKKAB;488 |
| 8 | 33444555669TJJQKB;345788899TJJQQQKS;3667789TKKAAAA222;7T2 |
| 9 | 3447889TJQKKAA222;333445669TTTQQQAA;5556677899JJJK2SB;78K |
| 10 | 35556JJJQQQKAA22S;44566677788999A22;3344789TTTTJQKKAB;38K |

Table 9: Game process while DR is Landlord

| NUMBER | GAME PROCESS |
|---|---|
| 1 | 0,33;1,TT;0,QQ;2,KK;2,J;0,K;1,2;1,56789;1,44;2,TT;2,Q;0,A;1,2;1,K;1,AAA3; 0,SB;0,JJJ66;0,456789;0,9 |
| 2 | 0,4443;1,5557;0,AAA5;0,6;1,8;2,2;2,3339;2,66;0,KK;0,22;0,SB;0,89TJQ |
| 3 | 0,56789;1,789TJ;0,9TJQK;0,AAA3;0,22277;2,SB;2,89TJQKA;2,66644;2,33;2,Q |
| 4 | 0,34567;1,789TJ;1,8;2,Q;0,2;2,S;2,4445553J;2,77;0,99;1,QQ;2,AA;2,88;0,JJ; 1,AA;0,KKKK;0,TTT3;0,B;0,Q |
| 5 | 0,33;1,99;2,QQ;0,AA;2,22;2,55667788;2,3;0,9;2,T;0,Q;2,A;0,2;1,S;0,B; 0,789TJQK;0,444TT |
| 6 | 0,3456789;2,456789T;2,3339;0,JJJ5;2,AAAK;0,222Q;0,K;1,2;1,44;2,QQ |
| 7 | 0,4;1,9;2,K;0,2;2,B;2,34567;2,9TJQKA;2,33;0,AA;1,22;1,44;2,JJ |
| 8 | 0,4445553366;0,7;1,S;0,B;0,T;2,2;2,6789T;0,9TJQK;0,2;2,AAAA;2,KK;2,7;0,J |
| 9 | 0,44;1,66;2,77;0,88;1,AA;1,3335;2,5558;0,KKK3;0,789TJQ;0,AA;0,2227 |
| 10 | 0,5556;1,9995;0,JJJ8;0,33;1,22;1,6667774488;1,A |

Table 10: Game process while human is Landlord

| NUMBER | GAME PROCESS |
|---|---|
| 1 | 0,456789;1,56789T;1,3;2,2;2,55;0,66;2,77;0,QQ;1,22;1,T;2,K;0,A;1,2;0,S;0,K;1,A;0,B;0,JJJ33;0,9 |
| 2 | 0,3;1,J;2,2;2,6789T;0,89TJQ;1,TJQKA;1,456789;1,7;2,Q;0,2;0,2;0,4445;0,AAA6;0,SB;0,KK |
| 3 | 0,3;1,7;2,Q;0,2;0,56789;0,9TJQK;2,TJQKA;2,66633;0,AAA77;0,22;2,SB;2,44;1,88;1,44;1,TT;1,JJ;1,KK;1,5553;1,2;1,9 |
| 4 | 0,3;1,8;2,Q;0,2;2,S;2,4445553J;2,77;0,99;1,QQ;2,AA;0,KKKK;0,Q;1,2;0,B |
| 5 | 0,9;1,K;2,A;2,55667788;2,QQ;2,KK;0,AA;2,22;2,3;0,2;1,S;0,B;0,33;1,99;0,TT;0,789TJQK |
| 6 | 0,Q;1,2;1,44;2,QQ;2,456789T;2,3339;0,JJJ5;2,AAAK |
| 7 | 0,4;1,9;2,K;0,A;1,2;0,S;2,B;2,34567;0,TJQKA;0,6;1,Q;0,K;1,A;0,2;0,7;2,A;0,2;0,99;0,8888;0,Q |
| 8 | 0,4445553366;0,7;1,9;2,2;2,6789T;0,9TJQK;0,T;1,J;2,2;0,B;2,AAAA;2,3;0,J;1,S;1,88;1,3;2,2;2,K;0,2 |
| 9 | 0,789TJQ;0,44;1,66;2,99;0,AA;0,KKK3;0,22288;2,SB;2,5558;1,TTT5;1,3339;2,JJJK;1,QQQ4 |
| 10 | 0,JJJQQQ68;0,55533;1,99944;1,6667775A;1,88;0,KK;1,22 |

