# OpenReview forum: "A Solution to China Competitive Poker Using Deep Learning"
_ICLR.cc/2019/Conference_

### Official Review · AnonReviewer2 · 2018-11-06
**promising performance, left more mysteries than observations**

**Rating:** 2
**Confidence:** 3

**Review:**

The authors propose a model that learns to play the China Competitive Poker game. The model uses CNN to predict the actions, and is trained from actual human game records. The model is shown to beat the current best AI and human amateur players.

The performance is certainly strong (if it were true). But given the double-blinded policy, there is literally no way to verify the correctness of the performance---in other words, the paper is currently not reproducible at all. So the following comments are based on the trust-worthiness of the paper.

(1) immature writing: The writing lacks formality and looks like a final project report. For instance, the super-short Section 2 is rather unprofessional---it is hard to believe that the related works can be described within two paragraphs anyway. Even as someone who understands the game of China Competitive Poker, I find it very hard to follow Section 3. There is a big room for improving the English writing.

(2) ill-illustrated specialty of the model: In particular, it is not clear why the model should be superior than other modeling choices. For instance, what role does the neighboring connections of CNNs play? What are the cons and pros of choosing CNNs? Are there strong motivations to design the model this way?

(3) many unanswered mysteries: why does the model trained with human records readily super-human? Note that this is controversial to common imitation learning where the typical performance is bound by the human-level performance. Even though authors claimed in the response that there are "many professional records"---but how many is many? Did the authors analyze the records and separate the professional versus amateur ones?

---

> ### Author Response · Authors · 2018-11-09
> **Thanks for your review.**
>
> Thanks for your review.
>
> (1)in other words, the paper is currently not reproducible at all. So the following comments are based on the trust-worthiness of the paper.
>
> We will offer a online interface to public, it will return a action when you sent cards and game process. You can test the model as you wish. It will take a few days to prepare I think. And, I can offer the video of test match (all 10 games) and public email addresses of four top amateur players, you can check with them.
>
> (2)immature writing: The writing lacks formality and looks like a final project report. For instance, the super-short Section 2 is rather unprofessional---it is hard to believe that the related works can be described within two paragraphs anyway. Even as someone who understands the game of China Competitive Poker, I find it very hard to follow Section 3. There is a big room for improving the English writing.
>
> I am really sorry about it. I will continue to optimize the paper and improve English.
>
> (3)ill-illustrated specialty of the model: In particular, it is not clear why the model should be superior than other modeling choices. For instance, what role does the neighboring connections of CNNs play? What are the cons and pros of choosing CNNs? Are there strong motivations to design the model this way?
>
> The following is the reason written in the paper:
>
> We choose CNN to solve the problem in CCP due to the following reasons: First, CNN has achieved superhuman performance in perfect information games. Second, there is semi-translational invariance in CCP, e.g. there are two sets of cards in the same category but with different ranks (like “34567” and “45678”, see more information in section 3), if we add each card’s rank, “34567” become “45678”, this is translational invariance. The player can play out “45678” after another one played out “34567”, but it is illegal if we swap the order, this is the reason for “semi”.
>
> Besides X-axis, I think there is translation invariance in Z-axis also. CNN can get good performance dealing with translation invariance.
>
> (4)many unanswered mysteries: why does the model trained with human records readily super-human? Note that this is controversial to common imitation learning where the typical performance is bound by the human-level performance. Even though authors claimed in the response that there are "many professional records"---but how many is many? Did the authors analyze the records and separate the professional versus amateur ones?
>
> First, many authors proved that the neural network can get the top amateur level in games just by supervised learning, like Chris J. Maddison’s paper “MOVE EVALUATION IN GO USING DEEP CONVOLUTIONAL NEURAL NETWORKS” showed: “We train a large 12-layer convolutional neural network by supervised learning from a database of human professional games. The network correctly predicts the expert move in 55% of positions, equalling the accuracy of a 6 dan human player.” 6 dan is top level in amateur players. Deepmind got similar conclusion in their paper.
>
> Second, compared to Go, there are seldom professional players in the strict sense in CCP. It is not a full time job for "CCP professional players", because players can not get enough money in CCP match. Maybe "semi-professional player" is more suitable, but they are really top players than others. As I said in previous reply, game records came from online platform. Players only need to offer a cellphone number or a tencent account to online platform. I think it is really very hard to distinguish just by them. many online platforms support for visitors to log in.

---

> > ### Comment · AnonReviewer2 · 2018-11-09
> > **thanks for clarifying**
> >
> > Thanks to the authors for clarifying. (1) and (2) are facts that cannot be disputed anyway, and I still hold my conservation on (3) and (4) because
> >
> > (3) While what the authors claim are weak reasons/conjectures of using CNN, those reasons are not strong. Or at least the paper fails to answer this question: "Does CNN perform well *because* it matches the translation variance needed for this task?"
> >
> > (4) What the authors claimed only shows that the network can *predict* how human moves well, but it does not explain why the network can *act* better than human. The fact that the data set contains possibly more low-level players than high-level ones makes it really mysterious on why the network can *act* better than all those players. The mystery is not well-answered by the authors' hand-waving statements---more scientific evidence is needed.

---

> > > ### Author Response · Authors · 2018-11-16
> > > **Thanks for your questions. We will optimize paper.**
> > >
> > > (3) While what the authors claim are weak reasons/conjectures of using CNN, those reasons are not strong. Or at least the paper fails to answer this question: "Does CNN perform well *because* it matches the translation variance needed for this task?"
> > >
> > > The answer is yes. We tried DNN, RNN, LSTM in CCP, did not achieve good performance. But we didn't record too much data.
> > >
> > > (4) What the authors claimed only shows that the network can *predict* how human moves well, but it does not explain why the network can *act* better than human. The fact that the data set contains possibly more low-level players than high-level ones makes it really mysterious on why the network can *act* better than all those players. The mystery is not well-answered by the authors' hand-waving statements---more scientific evidence is needed.
> > >
> > > I think human is a board word, and “*act* better than all those players” is not accurate. There are high level players records in data, although it's really hard to analysis.

---

> > > > ### Comment · AnonReviewer2 · 2018-11-16
> > > > **my follow-up two cents**
> > > >
> > > > (3) The authors' "yes" answer, to me, does not contain enough scientific proof. CNN has lots of difference to, say, a fully-connected network. Without a careful scientific study, it is not logical to attribute the success of the model to "translation invariance." For instance, CNNs' translation invariance is often attributed to pooling. Is there a comparison between the same CNN with and without pooling in order to justify that translation invariance is a key benefit in this task?
> > > >
> > > > (4) So now we are saying that the model learns from the top players to beat the good ones. While it sounds a bit more possible, it is still quite mysterious as (I assume) there are many more bad players than top ones. Any reasonable machine learning model would then be "biased" to mimic the bad players than the top ones. So the observation that your model beat the good ones (with your explanation of learning from the "top ones") still sounds suspicious.

---

### Official Review · AnonReviewer3 · 2018-11-08
**Good performance results, but not much scientific contribution**

**Rating:** 3
**Confidence:** 4

**Review:**

This paper provides a system to play CCP using some deep learning. The system consists of  three modules - the bid module, which is rule based, and the policy and kicker networks, which are simple convolutional neural networks. The authors use a dataset of 8 million game records consisting of 80 million state action pairs, and train the network in a supervised fashion. The resulting model is able beat MicroWe, the current state of the art in playing CCP, and even are able to beat a few "top amateur players"

- Why is the bid module also not learned? It seems like the feature set for the bid module is fairly simple, and a linear or MLP can do fairly well compared to a rule based module.
- It's not clear that separating the policy and kicker networks would be more advantageous than combining them. Thousands of actions is not a too large number - language modeling work routinely deals with outputting many more classes than that.
- Were the convolutions chosen 1D, 2D, or 3D? The figure seems to imply that the convolutions were over the XZ dimensions, with Y as the channel dimension. If so, this doesn't make too much sense to be, since the Z dimension is not uniform - the last index is all unseen cards, which is significantly more than the middle indices of "what was played in this round". There shouldn't be a lot of translational invariance in the Z dimension. I'm also not convinced that translational invariance is helpful in the X dimension.
- There were no comparisons with baseline models or different model architectures. I would like to see some results on the same structure, but with an Linear model, MLP or LSTM across the time dimension, or search through different types of convolutional networks.
- What hyperparameters were searched through in the learning process?
- Missing citations for MicroWe being the best CCP AI, and citations for the accomplishments of the top amateur players.
- How far away are the top amateur players from professional players? Please provide some context on how far this system is from solving CCP.
- Fig 3, 4 should just say #of games instead of "iteration"

This paper shows that one choice for a supervised learning system on a CCP game database can achieve amateur level human play. It does not give insight to why the system was designed this way, why the model choices were made, and how good simpler baselines might be able to achieve. The paper is not clearly written enough, and does not provide enough scientific value to be accepted to the conference.

---

> ### Author Response · Authors · 2018-11-16
> **Thanks for your questions. We will optimize paper**
>
>  It's not clear that separating the policy and kicker networks would be more advantageous than combining them. Thousands of actions is not a too large number - language modeling work routinely deals with outputting many more classes than that.
>
> Thousands of actions is not a too large number, but extreme examples are hardly found in records. We separated the network in order to make the both are simple. We will try to combine the policy and kicker network in our plan.
>
> - There were no comparisons with baseline models or different model architectures. I would like to see some results on the same structure, but with an Linear model, MLP or LSTM across the time dimension, or search through different types of convolutional networks.
>
> We tried different models in CCP, like DNN, RNN, LSTM, but the results were not very well (CNN was the best model), and we did not collect enough data. We will add these in the next version.
>
> Thanks for useful questions, we will add these in the next version.

---

### Public Comment · (anonymous) · 2018-09-29
**Weak Human Player?**

In table 9 and 10, game number 1.

Cards: 334566789JJQQKASB;34456789TTKAAA222;345577889TTJQQKK2;69J
Game Process: 0,33;1,TT;0,QQ;2,KK;2,J;0,K;1,2;1,56789;1,44;2,TT;2,Q;0,A;1,2;1,K;1,AAA3;0,SB;0,JJJ66;0,456789;0,9

Player 1 has a good start hand, however, he/she played not well.
Specifically, when he/she is in active mode after playing out 2; obviously he/she should play 3456789 instead of 56789.

---

> ### Author Response · Authors · 2018-09-29
> **Although the action looks like strange, the player has his thinking.**
>
> I searched the data and found "56789" was played out by Guofeng Xie, who won the champion in offline match. I contacted him and the following is his reply:
> First of all, "3456789" is a obvious normal and good choice. but the reason for "56789" is based on (1) He was not sure who had the last "2", he can only play out solo after played out "3456789", Landlord had two jokers, and maybe another "2", the game would be under the control of Landlord, so Guofeng Xie did not choose the longer solo with chain. (2) "J" had not been seen yet, he worried about Landlord had trio with pair, it is defense-based consideration. It is also hard even "4KAAA22" left, because Landlord may seperate the jokers. (3) "4KAAA22" and "344KAAA22" have advantages and disadvantages respectively. The former has less sets, while the latter is more flexible.
>
> The following is author's reply:
> We recorded the process with video. I found Guofeng Xie considered for a long time at that moment. It was really hard to choose.
> Thank you for your comments, waiting for more communication!

---

> > ### Public Comment · (anonymous) · 2018-09-29
> > **About Resources used**
> >
> > Thanks for your patient reply.
> >
> > Just wonder, how many GPUs did you used for training policy network?

---

> > > ### Author Response · Authors · 2018-09-30
> > > **Only NVIDIA 1080TI * 1**
> > >
> > > We found it is OK for training time, so we did not pay more attention to parallelization.

---

### Public Comment · (anonymous) · 2018-09-29
**result confidence interval.**

If you don't show the bot exploitability.  Please show the confidence interval about your results like Libratus and Deepstack did.  Because of poker is a stochastic game, need lots of hands to reduce the variance of the result. If the number of the matches is too small, the result is not compelling.

---

> ### Author Response · Authors · 2018-09-30
> **Good question!**
>
> About bot exploitability.
> Generally speaking, I think DeepRocket works well. It does stange actions in certain specific situations. E.g. landlord got "345555679TTQKA22B", output "3" rather than "34567" (I think most human players choose it) in a certain version.
>
> About the confidence interval.
> We tested standard deviation of landlord's scores, each iteration is 1000 games.
> 7.08798955981
> 7.11032987983
> 6.75631193774
> 6.92882089536
> 6.85500080233
> 6.6713368975
>
> I agree with you, there are great fluctuations in poker game result, so we use the duplicate mode. We can offer the DeepRocket server interface in order to test, if necessary.
> (I would add more data if you think it is not enough.)
> Thank you for your comment, waiting for more communication!

---

> > ### Public Comment · (anonymous) · 2018-10-08
> > **train data**
> >
> > what are you training data come from? For top human player and amateur player record ?

---

> > > ### Author Response · Authors · 2018-10-09
> > > **Mixed Data.**
> > >
> > > The main part of data came from the online game platform, and we choose high quality records, like from senior rooms. It is hard to distinguish which is top or amateur.
> > > We also add some records from top players, but not too much, less than 3%.

---

### Public Comment · (anonymous) · 2018-09-29
**question about Duplicate mode.**

why both peasants are AI in Figure5? Why it has three table, player change the position in turn? BTW, does human know what the position is AI during playing poker.

---

> ### Author Response · Authors · 2018-09-29
> **Hi, here it the answer for your questions!**
>
> I changed the sequence in order to better answer your question.
>
> 1.Why it has three table, player change the position in turn?
> There are two tables in test match, also shown in Figure5, not three. Players do not change the positions in turn, but it is allowed that the players want to swap their position (role) after the end of the game, just between human players, although the situation did not occur in test match.
>
> supplementary: there is another important imformation in plane, it is the number of left cards of the player.
>
> 2. why both peasants are AI in Figure5?
> First of all, the three players are a team. The peasants must cooperate with each other, so it is better if peasants come from the same team. The peasants are DeepRocket in table2, while human players in table 1.
>
> 3.BTW, does human know what the position is AI during playing poker.
> Yes, all human players in test match know identity of each position in both tables clearly.
>
> In fact, there are more complex duplicate mode, each team has four players, which include bid process. Please contact us when the authors are public, if you have interested in.
> Thank you for your comment, waiting for more communication!

---

### Public Comment · (anonymous) · 2018-09-29
**insufficiency matches ?**

only 10 matches with the human is unpersuasive,  some doubts:
(1)  The same model for both Peasant and Landlord? I think the policies of these two rules are distinct.
(2) Does this model really show cooperation between the two Peasant? How to explain if it does?

---

> ### Author Response · Authors · 2018-09-30
> **Thank you for your question!**
>
> (1)  The same model for both Peasant and Landlord? I think the policies of these two rules are distinct.
> Yes, the same model. And you are right, the policies are distinct. An important thing I forgot to mention in paper is role information is in each plane in policy network, in order to distinguish whose action. I am sorry about it.
>
> (2) Does this model really show cooperation between the two Peasant? How to explain if it does?
> To a certain degree. you can see example in paper, and we can find more examples. But it is not strong enough. The model does not work well when a peasant is in active mode, and have to help the partner to be in active mode next round. I think the model learned coorperation from human record, because the up peasant is a important position (does action before landlord), many obviously samples were found in record which were different from normal actions. And the role information in input of the policy network made an important impact also.
>
> The question about insufficiency, I only showed the first test match result in paper, by the way, the test match is offline. We did enough test then focused on reinforcement and Monte Carlo method. We can offer the DeepRocket server interface in order to test, if necessary.
>
> Thank you for your comment, , waiting for more communication!

---

### Public Comment · (anonymous) · 2018-10-22
**is it a special CNN network？**

First of all, a good job.
you mentioned  a ten layers CNN network is used in policy network，I want to know does this network have any special features?   if there is a network graph inserted in paper will be more well. Do you consider to use resnet instead?  as in Alphago Zero

BTW, it better to provide a reference link for MicroWe.   there isn't any introduction about MicroWe in google.

---

> ### Author Response · Authors · 2018-10-22
> **Good suggestion.**
>
> (1)I want to know does this network have any special features?
> I am not sure about the meaning of special features. The major features are in Y axis, like Trio or Solo. Other than these, we add the round info, role info and the number of left cards of each player as features. Could you show a example please?
>
>  (2)if there is a network graph inserted in paper will be more well.
> Thanks for your suggestion. I think I will add it in next version. Actually, we found something interesting about strides, layers, filters and accuracy, but the paper is limited in 8 pages, I have to give up introducing it in this paper.
>
> Do you consider to use resnet instead?  as in Alphago Zero
> About resnet, we already tried it in CCP. Generally speaking, we did not find obvious difference of performance between CNN and resnet, it seems CNN is a litter better than resnet according on the experiment result, but it is not enough. We will pay more attention in resnet when we defeat the human professional player.
>
> BTW, it better to provide a reference link for MicroWe.   there isn't any introduction about MicroWe in google.
> The issue about MicroWe, first, unfortunately, the author of MicroWe have not published any paper, it is a commercial software, and most of customers are domestic, second, although the game is very popular in China, it is still hard to find more infos in english, also a english reference.
> the website of National Computer Games Tournament (NCGT): http://computergames.caai.cn/
> but it is still chinese website.
> I am sorry about it. I will ask the committee of NCGT for more english infos.

---

> > ### Public Comment · (anonymous) · 2018-10-23
> > **Hope your further progress**
> >
> > Thank your reply.
> > I check the NCGT, really little english information about MicroWe. More knowledge about MicroWe will help readers to approve Deeprocket's ability.
> >
> > Another question, for last rounds play out cards, it is easy map to Y axis. But for hand left cards, it can be dismantling to lots kinds of possible types, solo, pair, trio etc,    for example,  5555 can be map to 5,55,555,5555, and also maybe in solo chain, pair chain..,   if more cards, it would be very complex.  did you do this dismantling by your self?  how to map to Y axis?  the cards dismantling in this kind of  CPP poker game will be more complex than Texas Hold’em

---

> > > ### Author Response · Authors · 2018-10-23
> > > **Thank you.**
> > >
> > > Because MW is a commercial software, the company offers online interface for test. I could ask for interface, and then maybe you can test. Otherwise, really hard to find English information.
> > >
> > > About dismantling, you are right, we did it by ourselves. Mapping them to all corresponding features in Y axis. We code different functions using python. they match each feature and add to Y axis.

---

### Public Comment · (anonymous) · 2018-10-22
**question about the policy network input**


In policy net input  15*19*21 , "Y-axis represents the number of each rank (from 1 to 4), and features in CCP, such as solo, pair, trio, etc".    the 19 means  that  the 15 kinds of cards types in Table 3,  and plus  the number of each rank(1-4) ?  it is a little hard to understand how to construct the input details.   could you show a example?

---

> ### Author Response · Authors · 2018-10-23
> **Only a little difference with your description.**
>
> It is 13 kinds, not 15 kinds. you can treat Rocket as a kind of Bomb. It is also not necessary adding "PASS" to in Y-axis.
> The left 2 is some other information, like round, role, the number of left cards.
> It is too complex to show a example here. Please reply me if it is still hard to understand.

---

> > ### Public Comment · (anonymous) · 2018-10-23
> > **Thanks for the details**
> >
> > Thank you for further explanation. But for the mapping, there is still some confusion.
> > For example,  the landlord play out a pair of 3, "33" in the first round
> > I assume the mapping like below，
> >
> > X：rank = "3" with rank 1，
> > Y： card type = pair， num of rank = 2，role = landlord， round = 1，
> > Z： 17  most recent round for player 1
> >
> > for the X Z axis，it is clear. while for the Y axis,  "33" is with num of rank 2, meanwhile a pair,    so it isn't one-to-one mapping
> >
> > the binary state S[i][j][k], which i j k should be uniquely determined I think.  Maybe my entire assumption is wrong.

---

### Public Comment · (anonymous) · 2018-10-23
**MicroWe was the best CCP artificial intelligence before?**

You mentioned"MicroWe won the championship in University Computer Games Championship & National Computer
Games Tournament 2017. MicroWe was the best CCP artificial intelligence before the appearance
of DeepRocket."
As we know,there are so many Doudizhu competitions in China.
How can MicroWe represent the best CCP artificial intelligence?

---

> ### Author Response · Authors · 2018-10-23
> **Is there many Doudizhu competitions for AI in China?**
>
> There are really many many Doudizhu competitions in China, like 360, TuYou, Tencent, JJ and many more， but only open to human players. If you know many competitions for AI, please tell us.
>
> Another question about how to prove MW represents the best CCP AI.
> Besides MW won the champion in NCGT, MW have been used in many online Doudizhu game, and get the rank No1 of market share. If you still doubt about it, I can give more details. That is why we think MW is the best and chose it to be opponent.

---

> ### Author Response · Authors · 2018-10-23
> **BTW, I know many competitions for AI in the game of Go.**
>
> I am a amateur player of Go. Tencent held competition for AI many times. Besides AlphaGo, there are many famous AI in Go, like "JueYi" and "JinMao" (sorry, Chinese pronunciation) which are developed by Tencent, "XingZhen" developed by Tsinghua University.
> But I seldom heard Doudizhu competition for AI.

---

### Public Comment · (anonymous) · 2018-10-28
**May I ask how did you get the training data?**

I'm wondering about:
1) How did you get 8 million matches' records?
2) Why such ordinary samples could achieve so high-level?

---

> ### Author Response · Authors · 2018-10-29
> **Good question.**
>
> 1) How did you get 8 million matches' records?
> we cooperate with online platforms, and get records.
>
> 2) Why such ordinary samples could achieve so high-level?
> I guess the reason is it includes many records generated by professional players. Although there are so many competitions in China, they still play online game, and have accounts on many platforms.  Overall quality of records is also high quality I think.
>
> The test match is held for test the distance between amateur players and DeepRocket, the result surprised us. We pay attention to Monte Carlo method for defeating professional players after test match.

---

> > ### Public Comment · (anonymous) · 2018-11-09
> > **Dataset Availability**
> >
> > Is there any chance that you can make this dataset publicly available so that we could potentially reproduce some of the experiments?

---

### Public Comment · (anonymous) · 2018-10-29
**which network model is Your CNN based,  AlexNet or VGG net？**

Could you give us some introduction of the Policy CNN network？

---

> ### Author Response · Authors · 2018-10-30
> **I think it is more like VGG**
>
> 1. As paper shows, we used 1 full connection layer.
> 2. We did not use pooling.
> 3. The size is 3*3 or smaller. Because input is asymmetrical shape, we used 1*3 also.

---

> > ### Public Comment · (anonymous) · 2018-10-30
> > **Without pooling, more parameters will be trained  there?**
> >
> > Thank you
> > If without pooling，I guess there will be 15*19*512 connections before the full connection layer.
> > more parameters will be trained.

---

> > > ### Author Response · Authors · 2018-10-31
> > > **Yes, more parameters if full connection is 15*19*512**
> > >
> > > This is not our design, but I think it also can work very well.
> > > We made it simpler compared to your idea

---

### Public Comment · (anonymous) · 2018-12-10
**To find out more**

Can you give more contacts, with whom could be discussed the delails of the topic and future cooperation?

---

> ### Author Response · Authors · 2018-12-21
> **Thank you for your reply**
>
> If you are interested in, please send an email to liuzx@smzy.cc and zzf@smzy.cc

---

### Meta-Review · Area_Chair1 · 2018-12-14
**Not clear enough, lacking details**

**Confidence:** 4
**Recommendation:** Reject

**Metareview:**

The paper presents a CNN that is trained from human games to predict which actions to take for China Competitive Poker (Dou dizhu).

The paper is poorly written, not because of the English, but because it is hard to understand the details of the proposed solution: it is not straight-forward to reimplement a solution from the presentation in the paper. It lacks explanations for several design decisions. This is unfortunate, as the authors point out in the rebuttal that they actually did way more experiments that are presented in the paper. Moreover, the experimental results lack comparisons to baselines, ablations, so that the proposed solution could be evaluated fairly.

In its current state, this paper can not be accepted for presentation at ICLR 2019.